

# Virtual Integration of Satellite and In-situ Observation Networks (VISION) v1.0: In-Situ Observations Simulator

Maria R. Russo,[1,2] Sadie L. Bartholomew,[1,3] David Hassell,[1,3] Alex M. Mason,[2] Erica Neininger,[4] A. James Perman,[2] David AJ Sproson,[5] Duncan Watson-Parris,[6] and Nathan Luke Abraham.[1,2]

[1] National Centre for Atmospheric Science, UK
[2] Yusuf Hamied Department of Chemistry, University of Cambridge, Cambridge, UK
[3] Department of Meteorology, University of Reading, Reading, UK
[4] Met Office, Exeter, UK
[5] FAAM Airborne Laboratory, National Centre for Atmospheric Science, University of Leeds, UK
[6] Scripps Institution of Oceanography and Halıcıoğlu Data Science Institute, University of California San Diego, USA

*Correspondence to*: Maria R. Russo (maria.russo@ncas.ac.uk)

**Abstract.**

This work presents the first step in the development of the VISION toolkit, a set of python tools that allows for easy, efficient and more meaningful comparison between global atmospheric models and observational data. Whilst observational data and modelling capabilities are expanding in parallel, there are still barriers preventing these two data sources to be used in synergy. This arises from differences in spatial and temporal sampling between models and observational platforms: observational data from a research aircraft, for example, is sampled on specified flight trajectories at very high temporal resolution. Proper
comparison with model data requires generating, storing and handling a large amount of highly temporally resolved model files, resulting in a process which is data, labour, and time intensive. In this paper we focus on comparison between model data and in-situ observations (from aircrafts, ships, buoys, sondes etc.). A stand-alone code, In-Situ Observation simulator, or ISO_simulator in short, is described here: this software reads modelled variables and observational data files and outputs model data interpolated in space and time to match observations. This model data is then written to NetCDF files that can be efficiently
archived, due to their small sizes, and directly compared to observations. This method achieves a large reduction in the size of model data being produced for comparison with flight and other in-situ data. By interpolating global, gridded, hourly files onto observations locations, we reduce data output for a typical climate resolution run, from ~3 Gb per model variable per month to ~15 Mb per model variable per month (a 200 times reduction in data volume). The VISION toolkit is fast and easy to use, therefore enabling the exploitation of large observational datasets spanning decades, to be used for large scale model
evaluation. Although this code has been initially tested within the Unified Model (UM) framework, which is shared by the UK Earth System Model (UKESM), it was written as a flexible tool and it can be extended to work with other models.



## 1 Introduction

The importance of atmospheric observations from both in-situ and remote sensing platforms has been growing in the last few
decades, with data archives, such as the new NERC Environmental Data Service (EDS[1]), becoming a key infrasructure for the
storage, exchange and exploitation of data. The strategic importance of in-situ measurements was also highlighted by the recent
£49M NERC funding to maintain and re-equip the BAe-146 research aircraft of the FAAM Airborne Laboratory out to 2040.[2]
Advances in geophysical model developments and exascale computing have similarly led to an increase in the complexity of
models used for climate projections in international modelling projects, such as CMIP6; chemistry and aerosols components
are now routinely being included in a number of climate model simulations (Stevenson et al., 2020; Thornhill et al., 2021;
Griffiths et al., 2021). Comparing all these models with observations is vital to increase our confidence in their ability to
reproduce historical observations, understand existing biases and ultimately to improve their representation of the atmosphere.

A wide variety of observational datasets can be used for model evaluation; what makes such comparisons with model data
inherently difficult is the difference between the orderly model data (defined on a regular 3D grid and at regular time intervals)
and the unstructured observational data (with variable coverage in space and time). A large computational effort is required
for the handling and processing of gridded model data files into a format suitable for direct comparison with observations,
especially when the measurement location varies with time (e.g aircraft, ships, sondes etc.). In order to compare to
observational data with varying coordinates, model output must include hourly (or higher frequency) variables over a large
atmospheric domain. As well as being data-intensive, extracting hourly data from a tape archive is also time-intensive. This
leads to orders of magnitude more data being stored, and processed than is actually required, and a significant amount of
manpower and computer resources are spent to extract, read and interpolate model data in space and time onto desired
observation coordinates. Because of these issues, previous studies of comparison between models and in-situ observations
from aircrafts, are generally restricted to case studies over a limited number of campaigns (e.g. Kim et al., 2015; Anderson et
al., 2021), or compare model data with observed data independently of the time or location of measurement (e.g. Wang et al.,
2020).

A previous attempt at producing Unified Model data on flight tracks was made several years ago, by embedding a flight_track
routine (using Fortran programming language) within the UM-UKCA source code (Telford et al., 2013). However, using this
approach added a computational burden to the running of the UM-UKCA model and it was mainly intended for output of
chemical fields (model diagnostics related to some UM dynamical fields were not available within the UM-UKCA
subroutines). As a result, the flight_track routine used in Telford et al. (2013) was never ported to further versions of UM-
UKCA.

---

[1] https://eds.ukri.org
[2] https://www.faam.ac.uk/mid-life-upgrade/





In this paper we describe the first tool in the VISION toolkit: ISO_simulator.py. This code can be embedded into the model workflow or can optionally be used as a stand-alone code with existing model data (e.g. to process variables from existing simulations). When ISO_simulator is embedded in the model workflow it produces much smaller data files which can be easily archived and are ready to be used for direct comparison with observations.

This new tool allows for routine production of model data interpolated at the time and location of in-situ observational data.

This can enable the exploitation of large observational datasets, potentially spanning decades, to be used for large scale model evaluation. The code is being developed to read any CF compliant model data and so could provide a valuable tool for supporting expanded diagnostics in upcoming CMIP7 experiments.

In section 2 we describe the ISO_simulator code, including command line arguments, input-output files and code optimisation. In section 3 we show how ISO_simulator is embedded within the Unified Model workflow. In section 4 we

provide some example plots showing comparisons of UM-UKESM modelled ozone to measurements from the Cape Verde Atmospheric Observatory (CVAO) (Carpenter et al., 2024), ozone measurements from the TOAR ocean surface database (including data from cruise ships and buoys, similar to Lelieveld et al., 2004 and Kanaya et al., 2019)  and ozone measurements from the FAAM Airborne Laboratory (Smith et al., 2024).

## 2 Description of ISO_simulator.py

In order to run ISO_simulator.py the user will need access to python3.8 or higher, including CIS (Watson-Parris et al., 2016), cf-python (Hassell and Bartholomew, 2020) and Iris (Hattersley et al., 2023) APIs.

ISO_simulator.py performs the following steps:

1. Reads time and coordinates from observational files using CIS python libraries.
2. Reads all model variables from hourly files using cf-python libraries.

3. Co-locates model variables in space and time to the same time/location as the observations using CIS python libraries.
4. Writes monthly NetCDF files (Rew et al. 1989) containing model variables co-located onto flight tracks.

### 2.1 Input arguments

ISO_simulator.py requires a number of command line arguments which are shown in Table 1. The current version of

ISO_simulator was developed for use within the UM modelling framework and therefore some of the current command line arguments are UM specific. However, when interfacing ISO_simulator to different models, these command line arguments can be changed to reflect output data that is specific to each model.

A subparser argument, 'jobtype', is used to indicate whether the code is running within a model run-time workflow (if 'batch' is selected) or as a standalone postprocessing tool, e.g. on existing model data, (if 'postprocessing' is selected). These subparser

arguments also unlock specific conditional arguments: --archive_hourly can be used only if 'batch' is selected and --




select_stash can only be used if 'postprocessing' is selected. By default, when running in batch mode, all fields present in the output file being processed will be co-located to the observational locations.

| ARGUMENT | DESCRIPTION |
|---|---|
| **-i --inputdir** *Directory_in* | *Directory_in* is the full path to the directory containing hourly pp files |
| **-t --obsdir** *Directory_obs* | *Directory_obs* is the full path to the directory containing observational files |
| **-d --cycle_date** *YearMonth* | *YearMonth* is a six digit tag to identify the start time of the analysis (YYYYMM) |
| **-n --n_months** *N* | *N* is the number of months to process, including *YearMonth* (optional; default 1) |
| **-r --runid** *UM_jobid* | *UM_jobid* is the unique identifier associated to a UM integration |
| **-p --ppstream** *Single_char* | *Single_char* is a single character identifying the hourly UM data ppstream as defined in Rose, e.g. k |
| **-v --vertical_coord** *coord* | *coord* is the coordinate for vertical interpolation: air_pressure or altitude; (optional; default=altitude) |
| **-e --extra_file** | *extra_file* is the filename (including full path) of model orography file; (optional, only required for hybrid theta-height coordinated if vertical_coord=altitude; default=Directory_ft/orography.pp) |
| **-o --outdir** *Directory_out* | *Directory_out* is the location to write output NetCDF files (optional). If *batch* is selected, output files are always written to *Directory_in* and additionally copied to *Directory_out* if present. If *postprocessing* is selected, output files are written to the current directory (./) or to *Directory_out* if present) |
| *batch* | Indicates the python script is running within the model workflow |
|     **-a --archive_hourly** | *True* to keep model hourly files instead of deleting them (optional; default True) |
| *postprocessing* | Indicates the python script is running with existing model files |
|     **-s --select_stash** *Code* | *Code* is a list of space separated UM stashcodes (an integer) to be interpolated (optional; default = interpolate all variables in the file) |

**Table 1: Description of command line arguments used to run ISO_simulator.py**

**2.2 Required Input files**

Model input files can be supplied in NetCDF, UM pp and UM fieldsfile formats and must have a date tag in the filename (YYYYMMDD) to identify the date in the file. The ability to read different formats of model input files gives extra flexibility to the code as it allows to read other model data as well as UM data.

The interpolation code can use either air_pressure or altitude as the vertical coordinate for interpolation. If this is not specified it will use altitude by default. When using air_pressure as a vertical coordinate, model variables are output on selected pressure levels. Since the UM has a terrain following, hybrid height vertical coordinate system, we additionally need to output a Heaviside function that accounts for missing model data where a pressure level near the surface falls below the surface height





for that gridbox. Where data is valid, the Heaviside function has a value of 1, and a value of 0 otherwise. By dividing the model
field on pressure levels by this Heaviside function, the model data is correctly masked and missing data is assigned to invalid
gridpoints.

When using altitude as a vertical coordinate, because of the UM hybrid height coordinates, model variables are defined at
specified heights above the model surface; therefore, the model orography field has to be provided to correctly convert the
model hybrid height to altitude above sea level. The name and path of the orography file can be defined using the -e input
variable.

For observational data which is defined at the surface (ground measurements or ship/buoy data), a vertical coordinate is
generally not provided. In this case, ISO_simulator will use the model lowest level and interpolate in time, latitude and
longitude only.

As well as model files, input files containing information on the observational data coordinates are also required. These input
files should be in NetCDF format (Rew et al. 1989), all data should be organised in daily files, and each file must have a date
tag in the filename (YYYYMMDD) to identify the date of the measurement. As well as time and positional coordinates, an
optional string variable can be added to the observational input files to identify data belonging to a specific dataset or campaign;
this can help during analysis to subset relevant data, which is useful when comparing to several datasets/campaigns over a
number of decades. Existing observational data might require a degree of pre-processing to ensure files are in the right format
to be used by ISO_simulator; the extent and type of processing will vary depending on the format and structure of each
observational dataset.

## 2.3 Output files

The model data, co-located to the observational coordinates, is generated in NetCDF format files. ISO_simulator produces one
file per model variable per month. The size of these output files depends on the number and size of the observational files on
which the model data is co-located and therefore can vary each month.

## 2.4 Code Optimisation

There are several python libraries that can deal with reading and writing of large, gridded data files. The choice to use CIS
python libraries in ISO_simulator.py stems from their ability to handle ungridded data (such as data from ships, aircrafts etc.)
and the ease of performing co-location from gridded to ungridded data. However, initial tests showed that reading UM model
input files using CIS was significantly slower than reading the same file with Iris or cf-python; Table 2 shows reading times
for loading files in the following formats, UM fieldsfile, UM pp, and NetCDF, using different libraries. The tests were
performed on the JASMIN data analysis facility (Lawrence et al., 2012).






| N of fields in file | 36 | 36 | 36 | 36 | 1 | 1 |
|---|---|---|---|---|---|---|
| N of fields read | 36 | 1 | 36 | 1 | 1 | 1 |
| File format | UM fieldsfile | UM fieldsfile | UM pp file | UM pp file | UM pp file | Netcdf file |
| Field dimension | 192,144,52,24 | 192,144,52,24 | 192,144,52,24 | 192,144,52,24 | 192,144,52,24 | 192,144,85,120 |
| CIS | - | - | - | 498.6 ± 7.1 s | 16.5 ± 0.2 s | 11.2 ± 0.1 s |
| Iris | 229.1 ± 1.1 s | 108.8 ± 1.0 s | 255.5 ± 2.8 s | 134.8 ± 1.7 s | 6.8 ± 0.3 s | 10.6 ± 0.1 s |
| Iris + structured UM loading | 38.3 ± 0.8 s | 39.6 ± 4 s | 54.9 ± 3.8 s | 52.9 ± 0.6 s | 2.3 ± 0.3 s | 10.4 ± 0.1 s |
| cf-python | 12.8 ± 1.5 s | 5.6 ± 2.3 s | 14.0 ± 4.2 s | 1.3 ± 0.2 s | 2.6 ± 0.3 s | 7.0 ± 0.1 s |

**Table 2: Comparison of file reading times using CIS library (version: 1.7.4), Iris library (version: 3.1.0) and cf-python library (version: 3.13.0). The times in the table include reading the file and accessing the data numpy array (via a simple print statement) to avoid lazy loading. The numbers in the table are averages and standard error for reading each file several times on the JASMIN data analysis facility.**

Given that potentially many such files would need to be read in each model month, cf-python was chosen to read the model data. However, CIS and cf-python use very different data structures for the gridded variables they read. In order to overcome this problem, a python function was developed to convert the cf-python gridded data structure to the CIS gridded data structure. Since reading model data is the slowest step in ISO_simulator, we further optimized the code by only reading model output files for days for which an observational input file exists.

## 3 Embedding ISO_simulator within the UM model workflow

This section describes how our code is interfaced within the UM framework. The UM uses Rose configuration editor (Shin et al., 2018) and the Cylc workflow engine (Oliver et al., 2018), respectively as a graphical user interface (GUI) and to control the model simulation workflow. Rose is a system for creating, editing, and running application configurations and it is used as the GUI for the UM to configure input namelists. Cylc is a workflow engine that is used to schedule the various tasks needed to run an instance of the UM in the correct sequence: for example *atmos_main* runs the main UM code, *postproc* deals with data formatting and archiving and *housekeeping* deletes unnecessary files from the user workspace.

A new Rose application, *VISION_iso*, was created and inserted into the Cylc workflow between the model integration step (*atmos_main*) and the *postproc* step (see Figure 1). This new application includes an input namelist and calls ISO_simulator.py; the NetCDF output files, containing model data co-located to the observations, are then sent to the MASS tape archive during the *postproc* step.





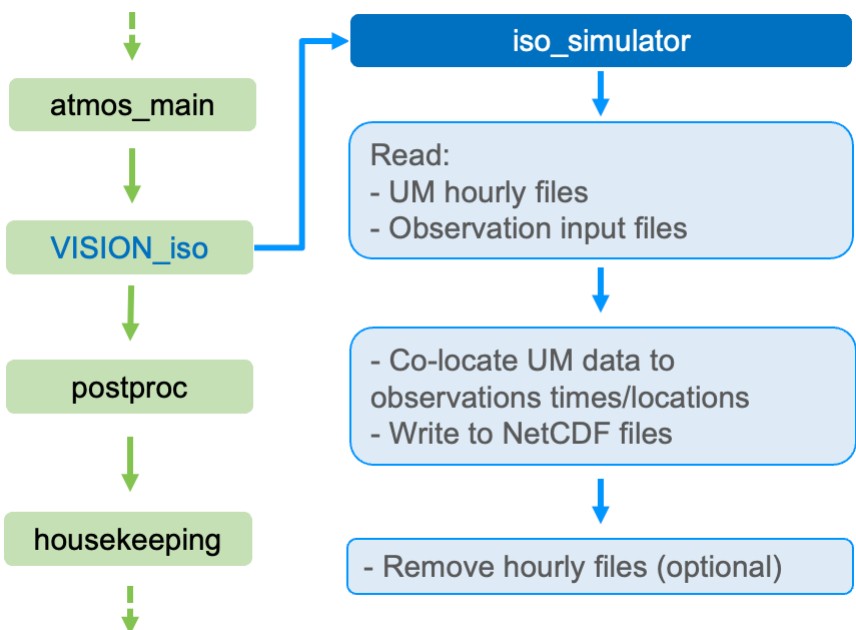

**Figure 1: A sketch of the UM run-time workflow, showing where the VISION toolkit is included.**

Since this software can be embedded into the UM run-time workflow and operates on UM output files (rather than being part of the UM source code), it has the following advantages compared to the approach in Telford et al. (2013):

1. Model data interpolated to the measurement times and locations is output using the internationally recognised NetCDF format, thus providing any required metadata information and making handling and analysis quicker and easier for users.
2. The code runs in parallel to the atmosphere model and does not affect the model run time.
3. The code can be easily customised to process any model data (not just UM data), therefore making it useful to the wider atmospheric science community.

Model data interpolated to the measurement times and locations can then be archived for long-term storage. When embedded into the UM workflow, data can either be transferred to the MASS tape archiving system or to the JASMIN data analysis facility (Lawrence et al., 2012). Further savings in data storage can be made by optionally deleting the hourly model output files used by the VISION toolkit.

**4 Test Cases**

In this section, we show some examples of using ISO_simulator to co-locate UKESM data to the same time and location as different types of observational datasets. Model simulations of UKESM were performed with a horizontal grid of $1.875^o \times 1.25^o$ and 85 vertical levels with a model top at 85 km and ERA reanalysis data was used to constrain the model





meteorology (Telford et al., 2008) to allow for better comparison with observations. For more information on the model configuration and details of the simulations, the reader is referred to the model description in Russo et al., (2023) and Archibald et al., (2024).

The aim of the plots in this section is not to answer specific science questions but to illustrate the way ISO_simulator can be used for comparison of model data with different sets of in-situ observations, namely: ground-based stations, ships/buoys, sondes/flights/UAV.

All model data is output hourly from UKESM, at a horizontal resolution of ~150km and then co-located, using ISO_simulator, to the same time and geographical coordinates as the observational data; the resulting data has the same time and spatial resolution as the observational data, making model data directly comparable to observational data. Furthermore, since the model and observational datasets can be compared over a long period of time, spanning several years, it is possible to sample seasonal and interannual variability, as well as better statistical sampling of extreme values. This type of comparison can greatly help to identify and improve model biases and to use models and observations in synergy to better understand atmospheric processes.

## 4.1 Cape Verde Atmospheric Observatory

The Cape Verde Atmospheric Observatory (CVAO) provides long-term ground-based observations in the tropical North Atlantic Ocean region (16° 51′ 49'' N, 24° 52′ 02'' W). The CVAO is a World Meteorological Organisation-Global Atmospheric Watch (WMO-GAW) station; measurements from CVAO are available from the UK Centre for Environmental Data Analysis (CEDA) data archives (http://catalogue.ceda.ac.uk/uuid/81693aad69409100b1b9a247b9ae75d5, Carpenter et al., 2024). The University of York provides the CVAO trace gas measurements, supported by the Natural Environmental Research Council (NERC) through the National Centre for Atmospheric Science (NCAS) Atmospheric Measurement & Observation Facility (AMOF). Data from CVAO was chosen as an example of surface station data because the ozone measurements are provided at a higher temporal resolution than the hourly model output; ISO_simulator can therefore be useful to interpolate model data in time to match the time of the observations.





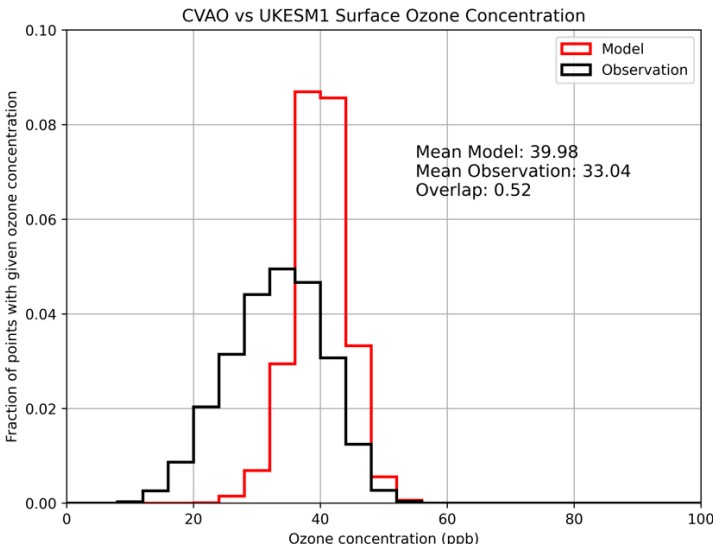

**Figure 2: Probability distribution of ozone concentration at CVAO between 2007 and 2022. The black line shows observed ozone and the red line is model data co-located using ISO_simulator.**


## 4.2 Ships and buoys dataset

The Tropospheric Ozone Assesment Report (TOAR[3]) is an international activity under the International Global Atmospheric Chemistry project. It aims to assess the global distribution and trends of tropospheric ozone and to provide data that are useful

for the analysis of ozone impacts on health, vegetation, and climate. A novel dataset has been produced by the TOAR "Ozone over the Oceans" working group. This dataset is an extension of previous similar datasets (Lelieveld et al., 2004, Kanaya et al., 2019) and it combines ship and buoys data from the 1970s to present day. This dataset will be released later this year as part of the TOAR-II assessment; given the large temporal span, this dataset constitutes a great example of using ISO_simulator to compare model and observational data over a large number of years.

UKESM data was co-located to the same times and locations as observations. The plot in Figure 3 shows the difference between modelled and observed ozone.

---

[3] https://igacproject.org/activities/TOAR/TOAR-II



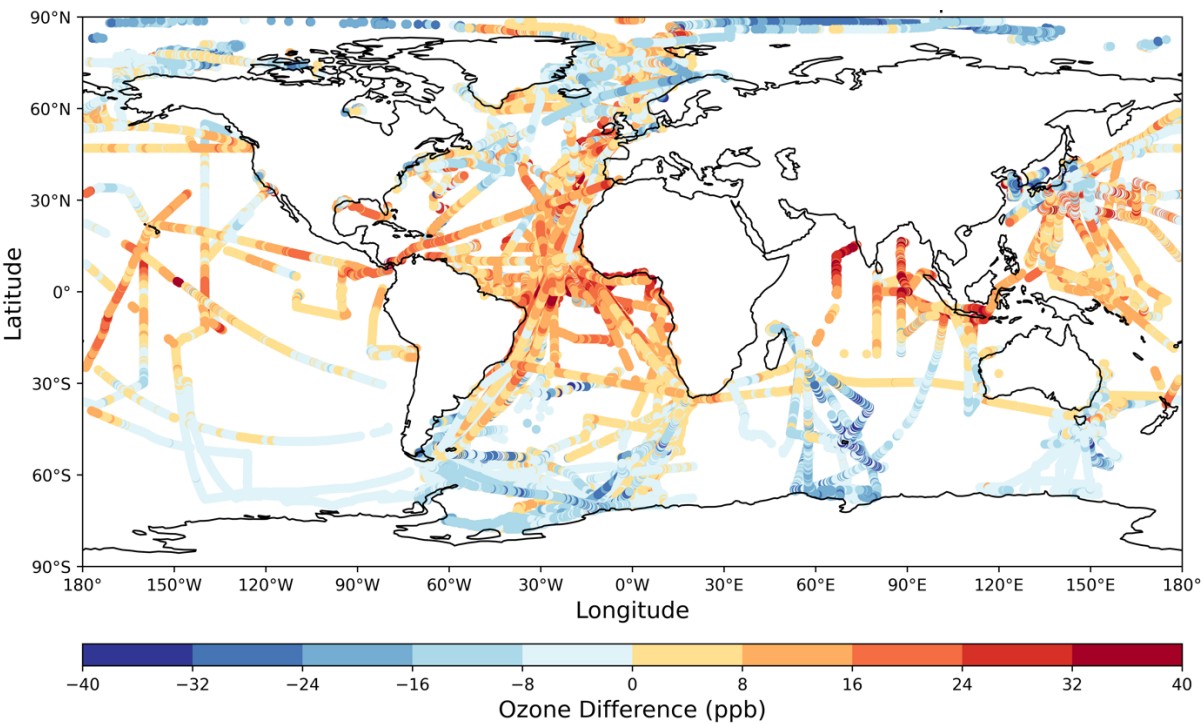

**Figure 3: Map plot difference between UKESM ozone and observed ozone for the period Jan 1986 to May 2022. Model data is co-**
**located using ISO_simulator.**

### 4.3 Aircraft data: comparison to FAAM and ATom aircraft

The FAAM Airborne Laboratory is a state of the art research facility dedicated to the advancement of atmospheric science.
It operates a specially adapted, BAE-146-301 research aircraft, and is based at Cranfield University. The FAAM Airborne
Laboratory is funded by the Natural Environment Research Council and managed through the National Centre for Atmospheric
Science (NCAS).

FAAM data from 2010 to 2020 was processed to extract ozone, time, latitude, longitude, air pressure and altitude and ensure
variable names were consistent throughout this time period (Russo et al., 2024a). Figure 4 and 5 show comparison of modelled
ozone and ozone observed by the FAAM aircraft. Figure 4 focuses on all individual flights from a specific campaign occurring
in Aug 2019. Figure 5 shows the difference between modelled and observed ozone for all flight points between the surface
and ~ 6 km and for all FAAM flights between 2010 and 2020.

The NASA Atmospheric Tomography (ATom) mission was a global-scale airborne campaign, funded through the NASA
Earth Venture Suborbital-2 (EVS-2) program to study the impact of human-produced air pollution on greenhouse gases and
on chemically reactive gases in the atmosphere. ATom utilized the fully instrumented NASA DC-8 research aircraft to measure
a wide range of chemical and meteorological parameters in the remote troposphere (Thompson et al., 2022). Data from the
ATom mission is available from the NASA data archive (https://doi.org/10.3334/ORNLDAAC/1925, Wofsy et al., 2021).





Figure 6 shows ozone concentrations from ATom and UKESM, as a function of time and altitude, for a specific flight on the 3rd of February 2017.

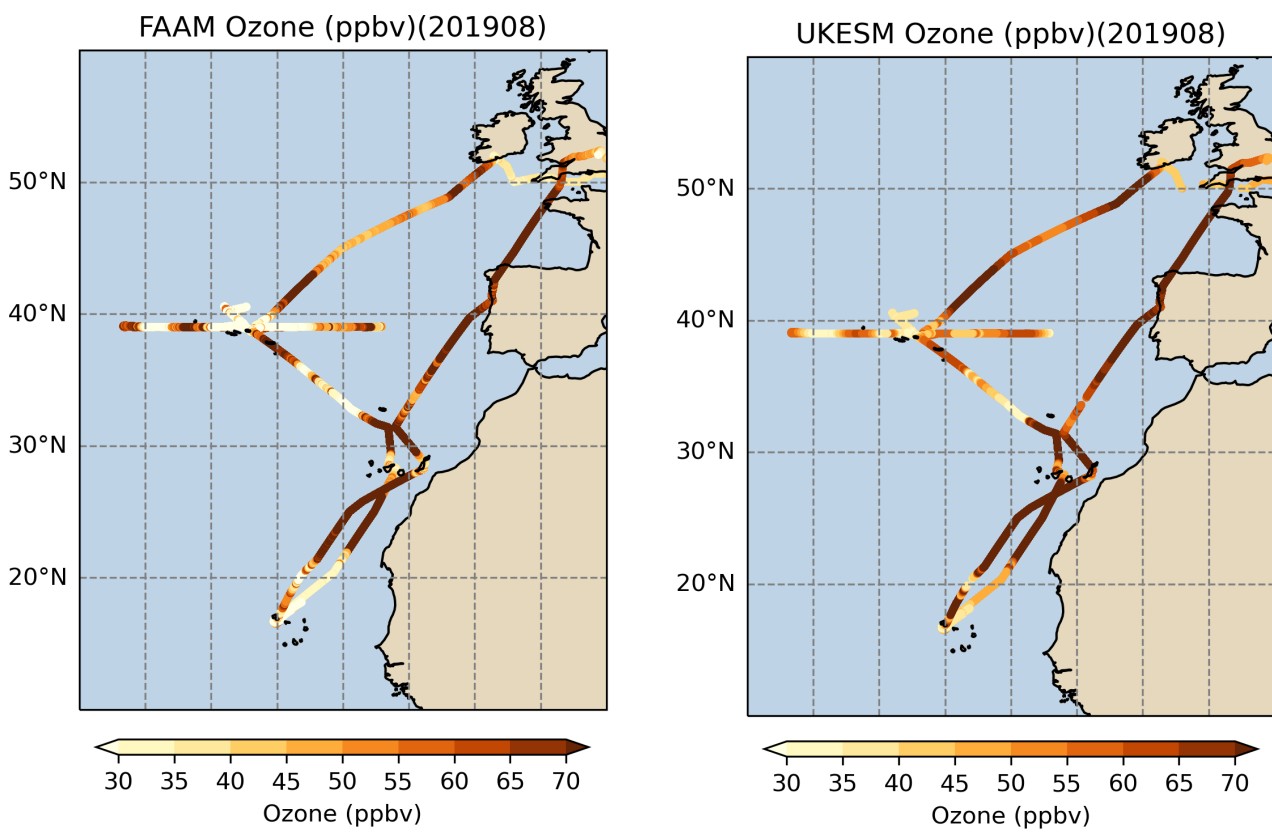

**Figure 4: FAAM and UKESM ozone concentrations (ppbv) for all flights in August 2019.**





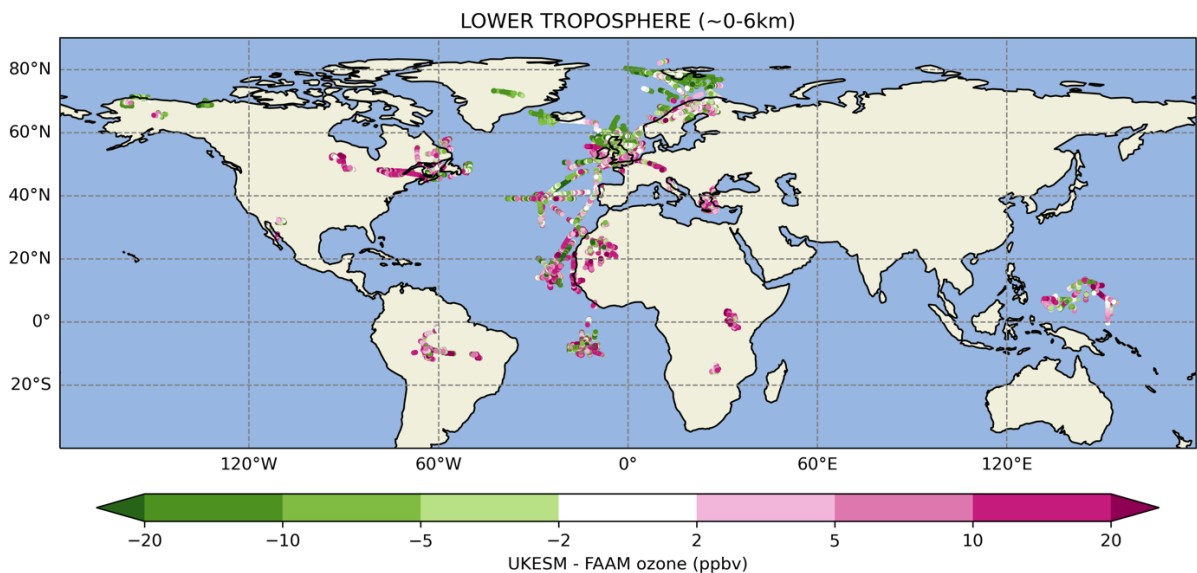

**Figure 5: Difference between UKESM and FAAM ozone concentrations (ppbv) in the lower troposphere (0-6km) for all FAAM flights which measured ozone between 2010 and 2020.**

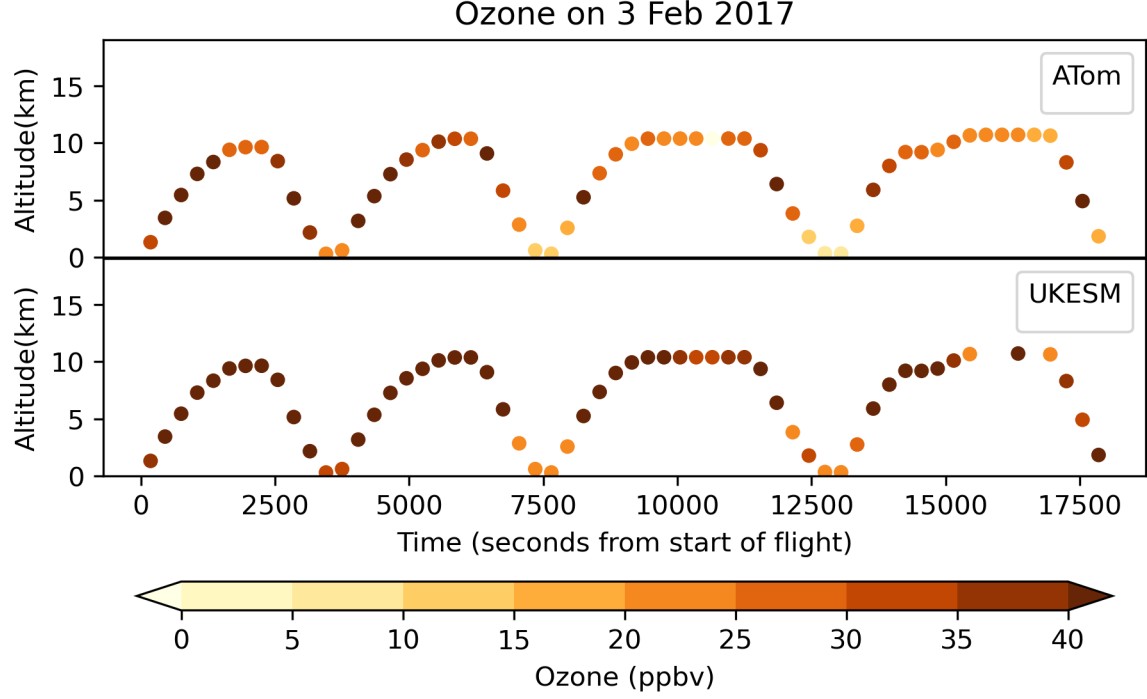


**Figure 6: ATom and UKESM ozone concentrations (ppbv) for the ATom flight on 3rd of February 2017 from Hawaii to Fiji.**



## 6 Conclusions

The ability to sample atmospheric model output at the same time and location as in-situ observations allows for better synergy
between model and observational data, resulting in better understanding of atmospheric processes and more effective model evaluation. However, to do this usually requires the processing of large volumes of high frequency gridded model data. By interfacing with the CIS and cf-python libraries, we are able to efficiently automate this step, greatly reducing manual post-processing time and the volume of data that needs to be saved following a model simulation. This method is also transferable to many different atmospheric models, and the code is provided on GitHub under an open-source license.

The use of the cf-python library to read-in the UM-format files significantly decreases the time taken to read these files when compared to the Iris or CIS libraries. This further reduces the time required on the HPC batch system needed to post-process the files from the global model to the times and locations of the in-situ observations. An extension to this work is currently being carried out to be able to output model data on satellite swaths for better comparison between atmospheric models and satellite data.

## 265 Code and Data availability

The current version of the code presented in the manuscript is available on GitHub https://github.com/NCAS-VISION/VISION-toolkit and archived on zenodo (Russo et al., 2024b) https://zenodo.org/records/10927302 under a BSD-3 license.

Input data are available from:
Modelled ozone (Abraham and Russo, last access Apr 2024),
https://catalogue.ceda.ac.uk/uuid/300046500aeb4af080337ff86ae8e776
FAAM ozone dataset (Russo et al., last access Apr 2024a),
https://catalogue.ceda.ac.uk/uuid/8df2e81dbfc2499983aa87781fb3fd5a
CVAO ozone dataset (Carpenter et al., last access Apr 2024)
https://catalogue.ceda.ac.uk/uuid/81693aad69409100b1b9a247b9ae75d5
ATom: Merged Atmospheric Chemistry, Trace Gases, and Aerosols, Version 2 (Wofsy et al., last access June 2024),
https://doi.org/10.3334/ORNLDAAC/1925

## Author Contribution

MRR developed the ISO_simulator code and processed the 11 year FAAM ozone dataset. MRR and NLA performed the UKESM model simulations. MRR, AMM and AJP performed data analysis and visualisation. NLA, EN, SLB, DH and DWP supported the development of the ISO_simulator code. DAS provided support with the FAAM data. NLA was the PI of the



projects leading to the development of ISO_simulator, supported by MRR, SLB and DH. MRR wrote the manuscript with contributions from all co-authors.

**Competing Interests**

The contact author has declared that none of the authors has any competing interests.

**Acknowledgements**

This work was funded by the Natural Environment Research Council (NERC) under the embedded CSE programme of the ARCHER2 UK National Supercomputing Service (http://www.archer2.ac.uk), hosted at the University of Edinburgh
(ARCHER2-eCSE02-2). This work used Monsoon2, a collaborative High-Performance Computing facility funded by the Met Office and NERC, the ARCHER2 UK National Supercomputing Service, and JASMIN, the UK collaborative data analysis facility. MRR, NLA, DH and SB are funded under the NERC VISION project (NA/Z503393/1), part of the NERC-TWINE programme.

We thank the Atmospheric Measurement & Observation Facility (AMOF), part of the National Centre for Atmospheric Science
(NCAS), for providing the CVAO dataset. We thank all contributors to the TOAR ship and buoy ozone dataset (Yugo Kanaya, James Johnson, Kenneth Aikin, Alfonso Saiz-Lopez, Theodore Koenig, Suzie Molloy, Anoop Mahajan, Junsu Gil). We thank the FAAM Airborne Laboratory for providing raw data input which was processed into the 11 year FAAM ozone dataset. We thank Ag Stephens and Wendy Garland from the Centre for Environmental Data Archival (CEDA) for technical support. We also thank Fiona O'Connor (UK Met Office) for suggesting the use of the CIS library.

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
