# Peer review of "Virtual Integration of Satellite and In-situ Observation Networks (VISION) v1.0: In-Situ Observations Simulator (ISO\_simulator)"

_Geoscientific Model Development, 2024_

## Author Comment (AC1)

The Authors want to take this opportunity to thank the reviewer for their time and their help in improving the manuscript.

Reviewer's 1 Comments

1. *On page 3, line 72 you speculate that this software could be a valuable tool for upcoming CMIP7 experiments. However, in the rest of the text, I get the impression that VISION is designed to work with hourly output from UM/UKESM with specific output file names. Furthermore, to my knowledge the most frequent global fields in CMIP are 3-hourly. Does the tool operate on CMIP6 CMORized output? If not, could you elaborate in the discussion what would be needed for VISION to claim a role in the CMIP7 assessment cycle?*

The VISION tools are currently capable of reading CMIP6 CMORized output using cf-python libraries. Whilst the UM/UKESM has been used as a test model and the current version of the tool is designed to work with UM output, the next version of the VISION toolkit (currently under development) will be model independent and is currently being tested with output from other models.

As part of the NERC-TWINE VISION project (which is partly funding the further development of the VISION toolkit), we aim to run training sessions and produce training material to support modellers who wish to integrate VISION into their model's workflow. If the CMIP community recognises the value of this tool and is willing to engage in the required training, some CMIP7 models could include VISION into their workflow and produce temporary hourly output which could be converted into CMORized co-located data for archiving.

In order to clarify this point, the text on page 3, line 72 will be changed from:

"The code is being developed to read any CF compliant model data and so could provide a valuable tool for supporting expanded diagnostics in upcoming CMIP7 experiments."

to:

"Whilst the UM/UKESM has been used as a test model and the current version of the tool is designed to work with UM output, the next version of the VISION toolkit (currently under development) will be model independent. Since VISION is designed to work with CF compliant data, including CMIP CMORized output, it could prove a valuable tool for supporting expanded diagnostics in upcoming CMIP7 experiments."

2. *In section 2.3 you briefly discuss the output of the VISION tool. Is the output CF-compliant? Please mention if so.*

The current version of the tool uses CIS libraries to write output in NetCDF format. Whilst this output can be easily read and processed, it is not fully CF compliant. The next version of the VISION toolkit (currently under development) is using cf-python libraries to write model output which will be fully CF compliant.

3.  *Table 2: you test different I/O libraries for NetCDF on performance. Surprisingly, the cf-python library is faster when reading a pp file with 36 fields than a single field. Could you elaborate on this?*

The JASMIN facility, on which tests described in Table 2 were performed, is a shared computing facility and the load on the system can vary at different times throughout the day. To avoid bias in the comparison of different libraries reading the same type of input file, all tests in the same column in Table 2 were performed simultaneously on the system. However, tests in different columns in Table 2 were performed at different times and this might result in perceived inconsistencies such as the one highlighted by the reviewer. Given that another reviewer has questioned the use of old version of the iris library, we have repeated these tests with the most recent versions of the three libraries and we have performed the tests on a local computer cluster to further minimise the impact of variable load on the timings in Table 2. This has resulted in more consistent numbers when looking at the performance of a Python library with multiple file formats.

> *In a broader perspective, I'm not sure whether a numpy.print is a good indicator for the I/O performance of the actual VISION workload, especially for distributed lazy I/O libraries under consideration. Maybe extracting values along a trajectory would provide a better indication of the performance? Please address this concern in the text.*

Initial timing tests using the VISION toolkit, identified reading of the model data as the single, most time-consuming step compared to reading of the observational data, extracting the values along a trajectory and writing the output. Therefore, we decided to focus our investigations on the time required to read the model data using different Python libraries, rather than the time required to read model and observational data and extract the value along a trajectory.  Since all the libraries used in the comparison store the data as a numpy array, using a numpy.print statement to access the variable data array was the simplest possible way to avoid lazy loading of the data.

To clarify this, the following sentence has been added to the manuscript on page 5, line 134:

"Initial timing tests using the VISION toolkit, identified reading of the model data as the single, most time-consuming step compared to reading of the observational data, extracting the values along a trajectory and writing the output. Therefore, the time required to read model data using different Python libraries was investigated."

4.  *Section 4: the examples only involve ozone concentration. Since the tool is presented as a general-purpose interpolator/collocator, one would expect multiple variables to be plotted for illustration.*

In this paper, ozone was chosen as an example because it is a widely measured chemical quantity with a large volume of data going further back in time. However, ISO_simulator can easily produce any modelled chemical and dynamical variable along specified tracks. To showcase this further, Fig 4 in the manuscript has been edited and the new figure contains two new panels with UKESM carbon monoxide and temperature, as well as UKESM ozone, along the FAAM flight track.

---

## Author Comment (AC2)

The Authors want to take this opportunity to thank the reviewer for their time and their help in improving the manuscript.

Reviewer's 2 Comments

> *(1) I believe that it would be worth mentioning the following tangential point. Another scenario where the overall approach described can be beneficial is for the creation of Nature Run (NR) simulations used for OSSEs. These will generally be much higher in resolution than climate simulations, and therefore allow for an even higher compression ratio. Further, practical limitations generally limit NR output to infrequent snapshots, whereas there is great research value in sampling at the model time step. However, to be practical, producing in-situ data in this configuration will generally require online processing to avoid the costly intermediate step of dumping full states to disk as the first step in the processing workflow. OTOH, online processing would allow for improved scalability of the interpolation step. It would be nice if future versions of VISION would allow for an online distributed interface for such scenarios.*

We agree with the reviewer that the VISION toolkit (including the new Satellite_simulator which is currently under development) could be of great benefit in a higher resolution Nature Run (NR) for Observing System Simulation Experiments (OSSEs). If properly integrated into the model's workflow, the in-situ and Satellite simulators from the VISION toolkit would allow full sampling of the NR simulation at the model timestep and it would achieve an even higher data compression ratio compared to its use with climate models. As part of the NERC-TWINE project that partly funds the further development of VISION, we are planning on simpler code (in the form of Python libraries) which will be easier to interface with a variety of different models and we also aim to provide training material and workshops to help users integrate these tools into their own model workflow.

To highlight the possible future application of VISION to the OSSEs problem, we have added the following sentence on page 3 line 71:

"Another possible application of the VISION toolkit is for improving model comparison with observations when conducting Observing System Simulation Experiments (OSSEs) (Zeng et al., 2020). These experiments are typically performed using models with a high spatial and time resolution; integrating the VISION tools into the workflow of such high resolution Nature Runs (NR) would allow to efficiently sample data at the model timestep with much reduced data storage requirements."

> *(2) To a limited degree a similar approach has been used for field campaigns. E.g.,*
> *https://github.com/GEOS-ESM/GMAOpyobs/blob/develop/src/pyobs/sampler.py*

The interpolation of data for better comparison of model and observations is not new, as evidenced in the introduction (line 53-56 and references therein). The VISION toolkit is relatively fast to run (takes less than 5 minutes to process 1 month of hourly model data) and can be automated to process large volumes of data at once, allowing for efficient data analysis over a large number of years.

---

## Author Comment (AC3)

Reviewer's 3 Comments

*Regarding section 2.4 "Code optimization": Could you give a rough estimate of how much real time running ISO_simulator takes? Are you using any of the parallel/distributed/out-of-core computing functionalities provided by cf-python or Iris via Dask? This could potentially lead to large performance improvements.*

ISO_simulator is a relatively fast code to run. Reading of the model data is the single, most time-consuming step; therefore the number of variables in the model hourly output file will also have an impact on the ISO_simulator run time. An estimate of the time required to run ISO_simulator is added at line 149 with the following sentence:

"In practical tests, when run over a large number of years, ISO_simulator takes around 2-3 minutes to process one variable for one model year."

VISION tools use Dask for reading of data with cf-python.

*Please consider publishing your code as a Python package on PyPI and/or conda-forge to enable installing it via "pip install " or "conda install ". This will greatly simplify the installation process, the dependency handling, and the inclusion of your code into other software. For example, other software products could simply do a "import <name_of_your_package" in their code.*

We understand the reviewer's point here. However, given that version 2 of the VISION toolkit is currently under development and will soon replace version 1, we  decided to include these recommendations on the next version of the code release, which will be provided as a Python package on PyPI as suggested and has already been minimally packaged such that it is installable locally from the open GitHub repository.

*Specific Comments*

1. *23: I think it would be helpful to also include the acronym "ISO_simulator" into the title. You mention it very often in the paper, so I think it deserves to be there.*

We have now added the acronym to the title.

2. *45: There are many models which also use unstructured grids (ICON, FESOM, etc.), so it's probably better to avoid the term "regular grid", which really is the opposite of an unstructured (or irregular) grid.*

We have now replaced the text in lines 44-46 from:

"what makes such comparisons with model data inherently difficult is the difference between the orderly model data (defined on a regular 3D grid and at regular time intervals) and the unstructured observational data (with variable coverage in space and time)."

to:

"what makes such comparisons with model data inherently difficult is the difference between the orderly model data, defined on the model grid at regular time intervals, and the unstructured observational data, with variable coverage in space and time."

3. *83-86: Mention what you need Iris for? Both other tools are mentioned here.*

This sentence has now been added at the end of line 81:

---

## Author Comment (AC4)

Reviewer's 3 Comments

> *Regarding section 2.4 "Code optimization": Could you give a rough estimate of how much real time running ISO_simulator takes? Are you using any of the parallel/distributed/out-of-core computing functionalities provided by cf-python or Iris via Dask? This could potentially lead to large performance improvements.*

ISO_simulator is a relatively fast code to run. Reading of the model data is the single, most time-consuming step; therefore the number of variables in the model hourly output file will also have an impact on the ISO_simulator run time. An estimate of the time required to run ISO_simulator is added at line 149 with the following sentence:

"In practical tests, when run over a large number of years, ISO_simulator takes around 2-3 minutes to process one variable for one model year."

VISION tools use Dask for reading of data with cf-python.

> *Please consider publishing your code as a Python package on PyPI and/or conda-forge to enable installing it via "pip install " or "conda install ". This will greatly simplify the installation process, the dependency handling, and the inclusion of your code into other software. For example, other software products could simply do a "import <name_of_your_package" in their code.*

We understand the reviewer's point here. However, given that version 2 of the VISION toolkit is currently under development and will soon replace version 1, we  decided to include these recommendations on the next version of the code release, which will be provided as a Python package on PyPI as suggested and has already been minimally packaged such that it is installable locally from the open GitHub repository.

> *Specific Comments*
>
> 1.  *23: I think it would be helpful to also include the acronym "ISO_simulator" into the title. You mention it very often in the paper, so I think it deserves to be there.*

We have now added the acronym to the title.

> 2.  *45: There are many models which also use unstructured grids (ICON, FESOM, etc.), so it's probably better to avoid the term "regular grid", which really is the opposite of an unstructured (or irregular) grid.*

We have now replaced the text in lines 44-46 from:

"what makes such comparisons with model data inherently difficult is the difference between the orderly model data (defined on a regular 3D grid and at regular time intervals) and the unstructured observational data (with variable coverage in space and time)."

to:

"what makes such comparisons with model data inherently difficult is the difference between the orderly model data, defined on the model grid at regular time intervals, and the unstructured observational data, with variable coverage in space and time."

> 3.  *83-86: Mention what you need Iris for? Both other tools are mentioned here.*

This sentence has now been added at the end of line 81:

"Iris libraries are used in some CIS functions to read gridded model data."

> 4. *Table 1: Please mention that input files can also be other formats than PP (like you do in the next paragraph).*

This has now been added.

> 5. *Table 2: Could you please explain what you mean by "Iris + structured UM loading" and why the difference is so big between "Iris" and that?*

The following sentence has been added to the caption in Table 2:

"The structured UM loading[1] method is a context manager which enables an alternative loading mechanism for 'structured' UM files, providing much faster load times."

With the footnote:

[1]https://scitools-iris.readthedocs.io/en/stable/generated/api/iris.fileformats.um.html#iris.fileformats.um.structured_um_loading

> 6. *143: Iris 3.1.0 is very old (Sep. 2021), have you considered using a later version?*

We agree with the referee and have decided that, for the purpose of Table 2, we will be using more recent versions of the same Python libraries, specifically CIS v1.7.9, Iris v3.10.0 and cf-python v3.16.2.

> 7. *189-191: You already mentioned a lot of this in the paragraph l.180-185, maybe you can unify this?*

We have now removed the sentence in line 180-181:

"In this section, we show some examples of using ISO_simulator to co-locate UKESM data to the same time and location as different types of observational datasets."

> *Technical Corrections*

> 1. *35: "NERC" is not defined*

Done

> 2. *114-115: "input variable" -> "command line argument"*

Done

> 3. *188: "UAV" is undefined*

Done